# What Affects Rural Ecological Environment Governance Efficiency? Evidence from China

**DOI:** 10.3390/ijerph19105925

**Published:** 2022-05-13

**Authors:** Min Qian, Zhenpeng Cheng, Zhengwen Wang, Dingyi Qi

**Affiliations:** 1School of Economics and Management, Wuhan University, Wuhan 430072, China; 2015101050004@whu.edu.cn; 2Department of Global Food Service, Woosuk University, Jeonju 55338, Korea; 3National Institute of Insurance Development, Wuhan University, Ningbo 315100, China; 4The Hotchkiss School, Lakeville, CT 06039, USA; dqi24@hotchkiss.org

**Keywords:** rural ecological environment, governance efficiency, influence factors, Tobit regression model

## Abstract

With rapid economic development, the protection of the ecological environment has become very important. The modernization of rural ecological governance is the basis and prerequisite for the sustainable economic and social development of vast rural areas of China in the current era. It is urgent to analyze the influencing factors and to improve China’s rural ecological environment governance efficiency for Rural Revitalization in the new era, and to promote the modernization of the national environmental governance system and governance capacity. This paper empirically examines the influencing factors on rural ecological environment governance efficiency in the whole country, and in the eastern, central and western regions separately, at the provincial level, using the Tobit regression model. The results show that, at the national scale, the level of rural economic development, the size of village committees and rural public participation all have positive roles in promoting the efficiency of rural ecological environment governance. Rural population agglomeration, financial support for agriculture. And environmental protection social organizations have negative roles, hindering the efficiency of rural ecological environment governance. From the perspective of the eastern, central, and western regions, the factors affecting the efficiency of rural ecological environment governance are different due to regional differences. According to the results of empirical analysis, it is proposed that the key issue in improving the efficiency of rural ecological environment governance in China is to promote differentiated regional coordinated governance mechanisms.

## 1. Introduction

As a public product, the ecological environment is the spatial foundation in which people live. A good ecological environment can not only provide people with a comfortable living environment to improve people’s sense of happiness and satisfaction, but also can promote people’s economic living standards and quality, which is the most inclusive aspect of people’s well-being. In recent years, with rapid economic development, the problem of ecological environment pollution has become more and more serious. Eco-environmental problems in China can be viewed from both urban and rural perspectives. The essence of the urban ecological environment problem is that the relationship between human beings living in the city and their living environment is unbalanced. As a developing country, China’s ecological and environmental problems have their own characteristics, such as poor air quality, severe acid rain leading to a prominent “heat island” phenomenon, and shortage of water resources; At present, China’s rural areas experience the following main problems: first, the unreasonable use of pesticides brings a series of environmental problems; second, the unreasonable use of chemical fertilizers brings a series of environmental problems; third, the burning of straw aggravates air pollution and the greenhouse effect. The prominence of rural ecological environment problems not only affects the improvement of the quality of life of urban and rural residents, but also affects the sustainable development of the entire society. In ecological environment protection, rural ecological environment protection should be given equal importance to promote the overall improvement of both urban and rural environmental quality. The governance of the rural ecological environment is not only the only means to ecological revitalization in the Rural Revitalization Strategy, but also the core meaning of building ecological civilization and Beautiful China and an important part of the modernization of the national environmental governance system and governance capacity [1,2]. At present, China’s rural ecological environment governance has gradually improved, but due to the impact of the urban-rural dual structure, the situation facing rural ecological environment governance is still serious. According to the announcement of the national survey of pollution sources in June 2020, rural pollution had become the main source of pollution by the end of 2017, and the pollution sources from agriculture are equivalent to the sum of industrial sources and domestic sources [3,4,5]. Due to the large amount of agricultural pollution emissions, the ecological environment is relatively fragile, and the resilience of recovery after damage is weak. In the process of modern economic development, more and more agricultural areas have been replaced by industrialization, and the ecological environment in rural areas has been seriously damaged. First, there is the problem of agricultural pollution emission amplification in the rural production environment. According to the data of the national environmental statistics bulletin, in 2015 the emission of COD from agricultural sources reached 10.686 million tons and the emission of ammonia nitrogen reached 726,000 tons, accounting for 48.1% and 31.6% of the total emission, respectively. The excessive use of chemical fertilizers, pesticides, and agricultural film has increased the pressure on the rural production environment [6,7,8]. Secondly, the foundation of rural human settlements is weak and its speed of improvement is slow. Rural infrastructure construction is the basis of the rural living environment, and domestic sewage treatment, domestic waste treatment, and toilet reconstruction are associated issues. Finally, the destruction of the rural natural environment is serious. Land desertification, deforestation, and other destructive acts are still not under control. The quantity and quality of natural resources such as cultivated land, forest land, grassland, and freshwater are gradually declining, the diversity of animals and plants in rural areas is sharply reduced, and the balance of the ecosystem is gradually damaged [9,10]. At present, it is an urgent task to improve the efficiency of rural ecological environment governance and to speed up the modernization of rural ecological governance capacity and its governance system.

Rural ecological governance aims to build a benign interactive governance method through the participation of grassroots governments, township enterprises and the general public, and to comprehensively control the destruction of natural resources and their environment, industrial enterprise pollution, agricultural non-point source pollution and livestock and poultry breeding pollution, as well as transforming and rectifying the deterioration of the rural living environment, and finally achieving the goal of improving the rural ecological environment and realizing the harmonious coexistence of man and nature. The modernization of rural ecological governance aims to use the power of scientific progress and environmental resources to drive rural economic and social development, and cooperate with corresponding industrial policies, fiscal policies, investment and financing policies and other means to strengthen environmental protection and ecological governance in rural areas, in order to ensure the coordinated development of the economy, society and ecological environment in rural areas. As the main factor in China’s agricultural production and farmers’ lives, the quality of the rural ecological environment is related to the health and social well-being of people all over the country. Promoting the efficiency of rural ecological environment governance is an important part of comprehensively promoting the construction of rural ecological civilization. The existing studies have more qualitative analysis on rural ecological environment governance, but less quantitative analysis on rural ecological environment governance efficiency and on analyzing the significant factors affecting rural ecological environment governance efficiency in different regions. Based on this, this paper attempts to analyze the influencing factors on rural ecological environment governance efficiency, taking the efficiency of rural ecological environment governance as the explained variable, with the level of rural economic development, rural population agglomeration, the scale of village committees, financial support for agriculture, environmental protection social organizations and rural public participation as explanatory variables. The Tobit regression model was used to analyze the influencing factors of rural ecological environment governance efficiency in the whole country and the eastern, central and western regions.

## 2. Literature Review

Adam Smith (1972) believed that an important indicator for evaluating efficiency was unit output, and believed that unit labor productivity could be improved through division of labor, thereby affecting the efficiency of the entire society [11]. At this time, economists’ thoughts about efficiency are mainly based on the core physical definition of efficiency. Generally speaking, efficiency is divided into both narrow and broad concepts. Efficiency in a narrow sense starts mainly from a micro perspective and refers to the operating efficiency of resources, how to minimize input, and maximize profits. Efficiency in a broad sense starts mainly from a macro perspective and refers to the optimal efficiency of resource allocation under certain input conditions. The efficiency of rural ecological environment governance is the balance of input elements and output elements in the process of rural ecological environment governance. Existing research does not have a clear and unified definition of the concept of rural ecological environment governance efficiency. Zhao Yuping (2019) believes that rural ecological governance efficiency aims to analyze the current rural ecological environment governance from the perspective of governance input and output. Significant achievements in the rural living environment, production environment, and natural resource environment are used to measure the effectiveness of the current rural ecological environment governance input elements [12]. According to the existing literature on the efficiency of rural ecological environment governance, and combined with the economic concept of efficiency, this study defines the efficiency of rural ecological environment governance as the process of rural green development, in which government and non-governmental entities synergistically utilize relevant resources to achieve efficient optimization of the input-output ratio of rural ecological environment governance.

At present, research on the governance of the rural ecological environment focuses mainly on the analysis of the relationship between agriculture and rural ecological environment. Since the 1990s, scholars have begun to pay attention to the application of mathematical models in research on rural ecological environment environments. Bekele et al. (2003), through the polynomial logit analysis of the survey data, showed that the holdings of people engaged in economic activities in each family harm decision-making related to water and soil conservation, and are closely related to the types of crops planted, the soil types of plots and the farming habits of farmers [13]. Reddy et al. (2006) analyzed the economic costs of water pollution and industrial pollution in rural areas based on detailed rural household data and proposed not only the passing of laws to improve the institutional structure but also sufficient governmental autonomy [14]. Hynes et al. (2014) believed that the policy orientation should take the correct protection of rural ecological environment resources as the main direction, and complete rural ecological environment protection through the management of the agricultural environment, farm management, and wildlife protection [15].

In terms of governance subjects, Michael (1951) first put forward the concept of multiple subjects in his book *The Logic of Freedom*. Scholars focus mainly on the major responsibility of the government. In the early stage, the responsibility orientation of the government was mainly focused on the level of law and policy formulation [16]. Ian (2001) believed that the government played a vital role in the development of agricultural environment-related policies, including public rights and interests, rural environmental capital investment, environmental policy and system formulation, and so on [17]. Since the 20th century, the government has put more emphasis on improving citizens’ awareness and on the spirit of environmental protection among its protagonists. The research of Mobin (2015) shows that the government should continuously improve the participation of the public and news media in environmental pollution control decision-making, in order to enhance the transparency of government decision-making [18].

Research on the countermeasures of rural ecological environment governance mainly focuses on tax and market regulation, total amount control, etc. Osborn et al. (2006) analyzed the advantages and disadvantages of the government’s supervision and non-supervision in the process of rural ecological environment governance and believed that the government should optimize policy tools to protect the rural ecological environment, including improving laws and regulations, tradable licenses, sewage charges, etc. [19] Bento Silva et al. (2015) assessed the relationship between the views of students in urban and rural communities around the Atlantic rainforest reserve in Pernambuco, Brazil, along with socio-economic factors. The results show that managers of environmental reserves need to promote meaningful interaction with student communities in rural and urban areas, improve the efficiency of these areas, and protect biodiversity [20].

Research in China on the efficiency of rural eco-environmental governance is still in its infancy. It mainly focuses on the measurement of rural eco-environmental governance efficiency based on cross-sectional data. Huang et al. (2015), based on the interprovincial panel data in 2011, used efficiency super-efficiency DEA to evaluate the efficiency of the input–output index system of rural ecological environment governance, constructed a comprehensive evaluation matrix combined with the level of rural economic development, and carried out cluster analysis on 31 provinces and regions [21]. Sun Yu et al. (2019) used the BCC model in the DEA method to construct the input–output index system of rural ecological environment governance; taking the cross-sectional data of 31 provinces in 2016 as the research object, they evaluated the efficiency of rural ecological environment governance in China [22].

In summary, there are still problems in the existing research on the efficiency of rural ecological environment governance: first, these studies only carried out research on rural ecological environment governance efficiency for a single year, and did not grasp the temporal and spatial evolution of China’s rural ecological environment governance efficiency; second, the research mainly discusses suggestions for improving the efficiency of rural ecological environment governance from the perspective of its redundancy rate and lacks discussion of its improvement from the perspective of its influencing factors. Therefore, this paper builds a scientific and comprehensive index system of rural ecological environment governance efficiency based on previous research, taking 30 provinces (autonomous regions and municipalities) in mainland China from 2007 to 2018 as the research object, using the Tobit regression model to analyze the regional differences in China’s rural ecological environment governance efficiency, in order to promote the overall improvement of the efficiency of China’s rural ecological environment governance. 

## 3. Index System Construction and Variables Selection

### 3.1. The Construction of an Evaluation Index System for Rural Ecological Environment Governance Efficiency

Referring to the practices of Huang and Sun [21,22], this paper launches the investment of rural ecological environment governance from three aspects: rural production environment governance investment, rural natural environment governance investment, and rural residential environment governance investment, and launches the output of rural ecological environment governance from three aspects: economic benefit, social benefit, and ecological benefit. The evaluation index system of rural ecological environment governance efficiency is constructed, as shown in Table 1.

The governance of the rural production environment mainly aims at the problems caused by the various natural and artificial transformations in the survival and development of agricultural organisms. The destruction of the rural production environment caused by natural disasters includes natural disasters such as floods, and waste generated in the process of agricultural growth [23,24,25,26,27,28]. The destruction of the rural production environment caused by man-made includes pollution caused by excessive use of chemical fertilizers and pesticides. Based on this, this paper selects the biogas treatment project for agricultural waste (10^2^ million yuan), drainage area (hm^2^), fertilizer applied application amount (10^4^ tons) and pesticide application amount (10^4^ tons) as the input indicators of rural ecological environment treatment. Rural living environment treatment mainly deals with the pollution of the rural living environment. Rural domestic garbage and rural domestic sewage are important factors affecting the rural living environment. The governance of rural human settlements mainly focuses on this, and governance is carried out by strengthening the investment of relevant funds, technologies, and other resources. Based on this, this paper selects a biogas digester for rural domestic sewage purification (item), rural domestic waste transfer stations (item), and rural environmental sanitation construction investment (10^2^ million yuan) as the investment indicators of rural residential environment treatment [29,30,31,32,33,34,35,36]. Rural natural environment management is mainly aimed at the environment, formed by natural features such as soil and water, biology, and climate. Rural natural environment governance mainly focuses on land degradation and biodiversity destruction. Based on this, this paper selects water and soil loss area (hm^2^), afforestation area (hm^2^), and rural landscaping construction investment (10^2^ million yuan) as the input index of the rural natural environment.

The economic benefit of rural ecological environment governance output refers to the output conducive to improving economic value during the process, conducive to providing a sustainable green production environment for crop production, in order to produce better ecological agricultural products and services. Based on this, this paper selects the output of green food, organic food, and pollution-free agricultural products (10^4^ tons) and the income of forestry tourism and leisure services (10^2^ million yuan) as the economic benefits of the output. The social benefits of rural ecological environment governance refer to the impact of improving people’s lifestyles and ideas in the process of rural ecological environment governance, to further improve the healthy living standard of villagers [37,38,39,40,41,42,43,44,45]. Based on this, this paper selects the popularization rate of sanitary toilets (%) and rural tap water (%) as the social benefits of output. The ecological benefit of rural ecological environment governance is to maximize the greening of the overall rural ecological environment and realize its green sustainable development and ecological livability. Based on this, this paper selects rural greening coverage (%) as the ecological benefit of output.

Data envelopment analysis is an efficiency evaluation method, based on multi-input and multi-output, to evaluate the relative effectiveness of objects. It was originally developed by American operations research scientist Charnes et al. on the basis of the concept of relative effectiveness. As the efficiency of rural ecological environment governance is an evaluation that needs comprehensive indicators, referring to the research of Zhang Jianqing et al., the DEA model is used to calculate the efficiency of rural ecological environment governance in China.

For the index layer in Table 1, due to the different dimensions of each index and its importance in the whole process of sustainable development, it is necessary to give different weights. This paper adopts the method of combining subjective and objective weights. The subjective weight is given by the analytic hierarchy process (AHP). The subjective weight is mainly based on the research of [46,47,48,49,50,51,52,53], and the Yaahp software is used for weighting. The objective weight is given by entropy method. The comprehensive weight is weighted by the D-S theoretical evidence synthesis method to avoid the limitation of using a simple average of subjective weight and objective weight. Through the sum of the product of standardized data and the weight of the indicators, we can arrive at the evaluation value of each primary indicator and incorporate the evaluation value into the DEA framework.

### 3.2. Variable Selection

The main purpose of this empirical study is to analyze the influencing factors affecting the efficiency of rural ecological environment governance in China, clarify the effect path of each influencing factor on the efficiency of rural ecological environment governance, and provide relevant policy suggestions for improving its efficiency. Based on the relevant polycentric governance theory, political economic theory, and previous research results [54,55,56,57,58,59,60], this paper focuses on the impact of the following factors on the governance efficiency of the rural ecological environment.

#### 3.2.1. The Level of Rural Economic Development and the Efficiency of Rural Ecological Environment Governance

The level of rural economic development has an important impact on the efficiency of rural ecological environment governance, which is mainly reflected in two stages. The first stage is that the development of the rural economy takes the rural ecological environment as the victim, demands too many resources, and produces a great deal of pollution and damage to the countryside, which hinders the efficiency of rural ecological environment governance. The second stage is to achieve coordinated development between rural economic development and the rural ecological environment. According to the Environmental Kuznets curve, when the economic development level of a region increases to a certain extent, the degree of environmental pollution will continue to decrease along with the continuous increase of people’s income [61,62,63,64]. The improvement of the rural economic development level has two positive effects on the efficiency of rural ecological environment governance: first, the improvement of the rural economic development level provides a solid economic foundation for rural ecological environment protection, which is conducive to safeguarding the environmental interests of the majority of villagers. Second, according to Maslow’s demand theory, with the rapid development of the rural economy, after meeting the most basic survival needs villagers begin to pursue higher-level needs, and the policy demand for rural ecological environment governance increases leading to improvements in the efficiency of rural ecological environment governance [65,66,67].

#### 3.2.2. Rural Population Agglomeration and Rural Ecological Environment Governance Efficiency

For rural areas, population aggregation will promote the continuous expansion of the rural scale. On the one hand, the increase in rural population will promote the rise of consumer demand, accelerate the consumption of resources and environment, and increase the difficulty of improving the efficiency of rural ecological environment governance. On the other hand, the increase of rural population is conducive to the centralized utilization of rural resources, improving the utilization efficiency of rural resources, and reducing the damage to the rural ecological environment. Rural population agglomeration is mainly manifested in the expansion of the rural population scale and the increase in rural population density. Different sizes of rural population and different degrees of rural population density have different effects on the efficiency of rural ecological environment governance [68,69,70,71,72].

#### 3.2.3. The Size of the Village Committee and the Efficiency of Rural Ecological Environment Governance

According to organic law, the village committee is a grass-roots mass autonomous organization, which manages the land and other property collectively owned by the farmers in the village, and is responsible for the public affairs and public welfare undertakings of the village. Therefore, the village committee not only needs to provide the management and protection of collective property in the village under its jurisdiction, but also needs to assume the role of agent for the public utilities in the village [73,74,75]. Rural ecological environment governance is an indispensable part of rural public utilities. The village committee plays a very important leading role in the efficiency of rural ecological environment governance, which lays a solid political foundation for improvement. In rural eco-environmental governance, generally speaking, the continuous expansion of the scale of the village committee is conducive to providing more policy support and resource preference for rural eco-environmental governance on the one hand, and convening more villagers and the public to actively participate in rural eco-environmental governance on the other. However, if the village committee deviates from the implementation of rural ecological environment governance policies, it will also directly affect the effect of rural ecological environment governance.

#### 3.2.4. Fiscal Expenditure on Agriculture and the Efficiency of Rural Ecological Environment Governance

Financial support for agriculture includes national financial support for agriculture, rural areas, and farmers, and is an important financial source for development. The impact of financial support for agriculture on the efficiency of rural ecological environment governance is mainly applied in two different ways: first, financial support for agriculture improves the solid economic foundation for rural ecological environment governance. Rural eco-environmental governance requires a great deal of investment, including human, capital, technology, and other resources. Solid financial support for agriculture is conducive to investing more resources in rural eco-environmental governance to improve its efficiency. The second is the rapid growth of financial support for agriculture, which can help the rapid economic development of rural areas, and to exchange the sacrifice of the rural ecological environment for the rapid growth of the rural economy. At the same time, the structure of financial support for agriculture guided by productive expenditure will further increase the damage to the rural ecological environment and bring great resistance to the governance of the rural ecological environment [76,77,78,79,80,81,82].

#### 3.2.5. The Social Organization of Environmental Protection and the Efficiency of Rural Ecological Environment Governance

Environmental protection social organization is non-profit social organization that provides environmental public welfare services for society. Its role orientation towards environmental protection is very important. Generally, environmental protection social organizations are divided into three types: environmental protection associations, environmental protection foundations, and environmental protection private enterprises. Environmental protection social organizations can have a positive impact on rural ecological environment governance. In terms of the expression of rural environmental interests, environmental protection organizations can have an active voice through various media to promote the implementation of rural environmental issues, and to promote the government’s application of corresponding measures. In terms of villagers’ interaction, we should make some rational expressions on behalf of villagers to effectively resolve rural environmental conflicts. Environmental protection social organizations represent the interests of the rural ecological environment, thus supervising and restricting the behavior of the government and enterprises [83,84,85,86,87,88]. At the same time, due to the limitations of the current local administrative system and the resources of environmental protection organizations, most environmental protection social organizations did not play their due role in the mass events occurring in the rural environment, and have even adopted silence and other coping strategies.

#### 3.2.6. Rural Public Participation and Rural Ecological Environment Governance Efficiency

American political scientist Easton [89] proposed that local government should fully consider the relevant opinions put forward by surrounding participants in the process of formulating public policies, to make certain targeted measures, and to put forward a system model: in the process of formulating policies, the public will put forward relevant opinions to local government according to their interest and demands. The input of the public and the local government together form a common political decision-making system. At the same time, local government will put forward practical policy plans according to the needs and participation of the public in order to form the output of the political decision-making system [90,91,92,93,94,95,96]. In the process of local government policy output, the public will have new expectations, which will further lead to new input and output. In the process of rural ecological environment governance, rural public participation is essentially a specific participation behavior in rural ecological environment governance, with the nature of supervision. Whether the role of rural public participation in rural ecological environment governance efficiency is hindered or promoted, and the size of the role, are uncertain. On the one hand, rural public participation is conducive to creating eco-environmental governance policies in line with the maximization of rural welfare effect, to enjoy various benefits brought about by the improvement in rural eco-environmental governance efficiency. On the other hand, the different degrees and methods of rural public participation will not only increase the cost and time of rural ecological environment management but also affect the science of rural ecological environment governance decision-making [97,98,99,100,101,102,103].

This paper selects rural economic development level indicators, rural population agglomeration indicators, village committee size indicators, financial support for agriculture indicators, environmental protection social organization indicators and rural public participation indicators as the core explanatory variables [104,105,106,107,108,109,110]. Among these, the rural economic development level is set by the actual per capita net income of rural residents (10^4^ yuan/person), and rural population agglomeration is set by the size of the rural population (100 million people) The size of the village committee is set based on the number of members of the village committee (10^4^ people), the financial support for agriculture is set based on the expenditure on agriculture, forestry and water affairs (10^4^ yuan), the number of ecological social groups (10^4^) is set by environmental protection social organizations, and the total number of letters regarding agricultural environmental pollution and ecological damage is set for rural public participation. The efficiency of rural eco-environmental governance in various provinces and cities is selected as the explanatory variable. Variable selection is shown in Table 2.

Among these, the rural population per capita, the level of rural economic protection and the level of rural public participation are selected as the indicators to explain Rural population agglomeration, which is set by the size of the rural population (10^3^ million person). The size of the village committee is set by the number of members of the village committee (10^4^ person), financial support for agriculture is set by the expenditure on agricultural, forestry and water affairs (10^4^ yuan), environmental protection social organizations is set by the number of ecological social groups (10^3^ items), and rural public participation is set by the total number of letters regarding agricultural environmental pollution and ecological damage.

### 3.3. Data Source

The time in which this research took place is 2007–2018. The basic data are from the 2008–2019 China Statistical Yearbook, China Rural Statistical Yearbook, China Economic Statistical Yearbook, China Environmental Statistical Yearbook, Provincial and Municipal Statistical Yearbook, and EPS database. The Chinese mainland’s 30 provinces (cities and districts) were selected as the research objects, and the data missing for Taiwan, Hongkong, Macao, and Tibet areas were significant and therefore not included in this study. In addition, a small amount of data cannot be directly calculated by the smoothing method in some provinces.

## 4. Empirical Analysis

### 4.1. Model

To avoid the error caused by the least square’s estimation, the restricted dependent variable model, the Tobit model, is usually used for regression analysis. Therefore, the Tobit regression model is used for analysis in this paper. The Tobit regression model is as follows:(1)$$E_{\mathrm{it}}=\alpha_{i}+\beta_{1}o_{it}+\beta_{2}p_{it}+\beta_{3}q_{it}+\beta_{4}r_{it}+\beta_{5}s_{it}+\beta_{6}w_{it}+\varepsilon_{i}$$

In Formula (1), $E_{\mathrm{it}}$ is the rural ecological environment governance efficiency of each province and city, $\alpha_{i}$ represents the constant term, $o_{it}$ represents the rural economic development level, $p_{it}$ represents the rural population agglomeration, $q_{it}$ represents the size of the village committee, $r_{it}$ represents the financial support for agriculture, $s_{it}$ represents the social organization of environmental protection, $w_{it}$ represents the rural public participation and represents the regression coefficient of each independent variable. *i* is the number of decision-making units, and $\varepsilon_{i}$ represents the error term of the regression equation.

### 4.2. Empirical Results and Analysis

Based on Stata 15.0, the regression results of the Tobit model are seen in Table 3.

It can be seen from the regression results that the level of rural economic development, the size of village committees, and rural public participation have positive roles in promoting the efficiency of rural ecological environment governance, they all pass the significance test of 1%, and their action coefficients are 0.253, 0.019 and 1.366, respectively. This means that for every 1% increase in the level of rural economic development, the size of village committees, or rural public participation, the efficiency of rural ecological environment governance will increase by 0.253%, 0.019%, and 1.366%.

According to the regression results, rural population agglomeration, financial support for agriculture, environmental protection, and social organizations all hurt the efficiency of rural ecological environment governance. Through the significance test of 1% or 5%, the action coefficients are −1.158, −0.141, −0.074. This means that every 1% increase in rural population agglomeration, financial support for agriculture, or environmental protection social organizations will reduce the efficiency of rural ecological environment governance by 1.158%, 0.141%, and 0.074%.

From the east, middle and west, the level of rural economic development hurts the efficiency of rural ecological environment governance in the eastern region. Through the significance test of 1%, the effect coefficient is −0.235, which means that for every 1% increase in the level of rural economic development, the efficiency of rural ecological environment governance will be reduced by 0.235%. The level of rural economic development has a positive role in promoting the efficiency of rural ecological environment governance in the central and western regions. They all pass the significance test of 1% or 5%, and their action coefficients are 0.527 and 0.731, respectively, which means that for every 1% increase in the level of rural economic development the efficiency of rural ecological environment governance will increase by 0.527% and 0.731%.

Rural population aggregation hurts the efficiency of rural ecological environment governance in the eastern region. Through the significance test of 1%, its effect coefficient is −0.235, which means that for every 1% increase in the level of rural economic development, the efficiency of rural ecological environment governance will be reduced by 0.235%. Rural population aggregation has not passed the significance test for rural ecological environment governance efficiency in the central and western regions, which shows that there is no statistically significant correlation there between rural population aggregation and rural ecological environment governance efficiency.

The size of the village committee has a positive role in promoting the efficiency of rural eco-environmental governance in the eastern and western regions. Both pass the significance test of 1%, and their action coefficients are 0.015 and 0.033, which means that for every 1% increase in the level of rural economic development the efficiency of rural eco-environmental governance will increase by 0.015% and 0.033%. The scale of the village committee has not passed the significance test for rural ecological environment governance efficiency in the central region, which shows that there is no statistically significant correlation between the scale of the village committee and rural ecological environment governance efficiency in the central and western regions.

Financial support for agriculture has a negative hindering effect on the efficiency of rural ecological environment governance in the central and western regions. They all pass the significance test of 1%, and their action coefficients are −0.340 and −0.386, which means that for every 1% increase in the level of rural economic development the efficiency of rural ecological environment governance will be reduced by 0.340%, and 0.386%. Financial support for agriculture has a positive role in promoting the governance efficiency of the rural ecological environment in the eastern region. Through the significance test of 1%, its action coefficient is 0.312, which means that for every 1% increase in the level of rural economic development, the governance efficiency of the rural ecological environment will increase by 0.312%.

The social organization of environmental protection hurts the efficiency of rural ecological environment governance in the central region. Through the 10% significance test, its effect coefficient is −0.140, which means that for every 1% increase in the social organization of environmental protection the efficiency of rural ecological environment governance will be reduced by 0.140%. The construction level of environmental protection social organizations did not pass the significance test on rural ecological environment governance efficiency in the eastern and western regions, which shows that there was no statistically significant correlation between environmental protection social organizations in the central and western regions and rural ecological environment governance efficiency.

Rural public participation plays a positive role in promoting the efficiency of rural ecological environment governance in Central China. Through the 5% significance test, its coefficient of action is 0.083, which means that for every 1% increase in the level of rural economic development the efficiency of rural ecological environment governance will increase by 0.083%. Rural public participation hurts the governance efficiency of the rural ecological environment in the eastern and western regions. Through the significance test of 1% or 5%, its action coefficient is −0.003 and −0.010, which means that for every 1% increase in the level of rural economic development the governance efficiency of rural ecological environment will be reduced by 0.003% and 0.010%.

### 4.3. Discussions

#### 4.3.1. The Analysis of the Influencing Factors on National Rural Ecological Environment Governance Efficiency

The level of rural economic development is conducive to improving the efficiency of rural ecological environment governance. The improvement of the rural economic development level means that resources, talents, science and technology, and other economic development elements are highly concentrated, which is conducive to the efficient utilization of resources, and the carrying capacity of resources and environment is continuously enhanced. With the development of the rural economy, China has provided a solid development foundation for rural ecological environment governance. After meeting the most basic needs, rural residents began to increase their demand for sustainable development such as protection of the environment, which has accelerated the process of rural ecological environment governance and improved its efficiency.

The size of the village committee is conducive to the improvement of the efficiency of rural ecological environment governance. With the continuous expansion of the scale of village committees in China, the level of handling public affairs in villages has been continuously improved. Not only is attention paid to the planning of village economic construction but also to the continuous improvement of the planning of ecological civilization construction: correctly managing rural resources and the environment, putting forward targeted ecological protection planning measures for ecological elements such as mountains, rivers, forests, fields, lakes, and village houses, putting forward targeted laws and regulations for the improvement of the rural residential environment, building a village ecological space system, and optimizing the governance of the rural ecological environment.

Rural public participation is conducive to the improvement of rural ecological environment governance efficiency. In the process of rural ecological governance in China, rural public participation, as a way of supervision, can not only quickly and effectively solve the problems which occur in the process of rural ecological environment governance, but also can realize the formation of a kind of supervision of the improvement of public consciousness in the process of rural ecological environment governance. At the same time, in the process of rural ecological environment governance, the Chinese government has taken into account the need for public participation in rural ecological environment governance according to local conditions, issued relevant policies, and invested a great deal of resources into rural ecological environment governance, to improve its efficiency.

Rural population agglomeration hinders the efficiency of rural ecological environment governance. Rural population agglomeration is the change of rural population quantity, structure, distribution, and migration. On the one hand, rural population agglomeration changes rural economic and social life, including production, consumption, culture, technology, and policy, and directly destroys resources and the environment. The threshold of environmental carrying capacity is becoming less and less, which increases the difficulty of rural ecological environment governance. On the other hand, with the increase in China’s rural population, governance impact on the rural ecological environment will achieve a multiplier effect, which will put great pressure on the potential of rural resources and environmental quality.

Financial support for agriculture hinders the efficiency of rural ecological environment governance. Although the total amount of financial support for agriculture in China is expanding, the bias of financial support for agricultural structures has a slow impact on the governance efficiency of the rural ecological environment [47]. For a long time, in the process of GDP development, the Chinese government has favored the concept of “rapid economic growth at the expense of the environment” and “pollution before treatment”, which makes the government focus on production expenditure in the structure of financial support for agriculture, and reduce rural ecological environment standards, with extensive and large-scale investment.

The social organization of environmental protection hinders the efficiency of rural ecological environment governance. It is oriented to maintaining the ecological environment, which usually promotes the work of ecological environment governance. However, at present, the social organization of environmental protection has a slow impact on the efficiency of rural ecological environment governance in China, which is due to the following reasons: on the one hand, China’s current environmental and social protection organizations have ignored various ecologically and environmentally damaging events in rural areas under the condition of their resource constraints in the current local administrative system. On the other hand, the work center of environmental protection social organizations is in cities. In the process of assisting cities in ecological environment governance, it may be contrary to the interests of the rural ecological environment, which increases the difficulty of rural ecological environment governance and reduces its efficiency [48].

#### 4.3.2. The Analysis of the Influencing Factors on Rural Ecological Environment Governance Efficiency in the Eastern, Middle and Western Regions

The level of rural economic development hurts the efficiency of rural ecological environment governance in the eastern region. The level of rural economic development in the eastern region is the fastest among the three regions. With the rapid development of the rural economy, the speed of the plundering of rural ecological resources and the environment is accelerating, resulting in a large number of point source and non-point source pollution, which puts great pressure on the rural ecological environment, thus reducing the efficiency of rural ecological environment treatment. The level of rural economic development has a positive role in promoting the efficiency of rural ecological environment governance in the central and western regions. Although the rural economic development level in the central and western regions is relatively low, in the process of rural economic development we pay attention to the coordinated development along with the rural ecological environment, which does not sacrifice the rural ecological environment excessively, and pay attention to the investment optimization of rural ecological environment governance, which promotes the improvement of its efficiency.

Rural population aggregation damages the efficiency of rural ecological environment governance in the eastern region. The impact of rural population agglomeration on the efficiency of rural ecological environment governance in eastern China is similar to that of the efficiency of rural ecological environment governance in China as a whole. Rural population aggregation has not passed the significance test for rural ecological environment governance efficiency in the central and western regions. The reason may be that a large number of rural people in the central and western regions have migrated to the eastern region, greatly slowing the plundering and destruction of rural ecological resources and environment in the central and western regions, which is conducive to rural ecological environment governance. However, it is the outflow of a large number of rural people that poses a serious threat to the resources of rural ecological environment governance. The outflow of human resources has caused a lack of other resources, which damages the efficiency of rural ecological environment governance. Therefore, in the central and western regions, the actual roles of rural population agglomeration and rural ecological environment governance efficiency are inseparable, which ultimately depends on other factors.

The size of the village committee has a positive role in promoting the efficiency of rural eco-environmental governance in the eastern and western regions. The effect mechanism of the size of village committee on the governance efficiency of the rural ecological environment in the eastern and western regions is similar to that of the size of the village committee on the governance efficiency of the rural ecological environment in the whole country. The scale of the village committee has not passed the significance test for rural ecological environment governance efficiency in the central region. The reason may be that the village committee faces the problem of policy implementation deviation in the process of dealing with public affairs in the village under its jurisdiction. It excessively focuses on other transactional work and does not put the rural ecological environment governance in a core position, so it is difficult to form an obvious role in rural ecological environment governance.

Financial support for agriculture has a negative hindering effect on the efficiency of rural ecological environment governance in the central and western regions. This negative effect of financial support is similar to that of financial support for agriculture on the efficiency of the rural ecological environment in China as whole. Financial support for agriculture has a positive role in promoting the governance efficiency of the rural ecological environment in the eastern region. The expenditure scale of financial support for agriculture in the eastern region has expanded continuously, which provides solid financial support for the governance of the rural ecological environment. At the same time, due to the optimization of the structure of financial support for agriculture in the eastern region, we pay attention not only to productive expenditure but also to rural constructive expenditure. We will continue to accelerate investment in rural ecological environment construction and improve the green and sustainable development of rural areas.

The social organization of environmental protection damages the efficiency of rural ecological environment governance in the central region. The negative barrier mechanism of environmental protection social organizations regarding the efficiency of rural ecological environment governance in Central China is similar to that of environmental protection social organizations in China as a whole. The construction level of environmental protection social organizations did not pass the significance test for rural ecological environment governance efficiency in the eastern and western regions. The reason may be that the environmental protection social organizations in the central and western regions focus on industrial pollution control in urban areas in the process of environmental governance. Due to the limitation of resources, the work of rural ecological environment governance is not carried out sufficiently, and therefore cannot form a substantive role in promoting rural ecological environment governance.

Rural public participation plays a positive role in promoting the efficiency of rural ecological environment governance in Central China. Rural public participation has a positive effect on the governance efficiency of rural ecological environment in Central China, and the mechanism is similar to that of rural public participation in China as a whole. Rural public participation damages the governance efficiency of the rural ecological environment in the eastern and western regions. The appeal to the rural public in the rural ecological environment governance has not been well met, which has an impact on scientific decision-making and is not conducive to the rural ecological environment governance.

## 5. Conclusions and Suggestions

In summary, this paper takes the efficiency of rural ecological environment governance as the explanatory variable, the level of rural economic development, rural population agglomeration, the size of village committees, financial support for agriculture, environmental protection social organizations, and rural public participation as the explanatory variables, and uses the Tobit regression model to analyze the influencing factors on rural ecological environment governance efficiency in the whole country, the eastern, central and western regions. Through empirical research, it is found that, at the national level, the level of rural economic development, the size of village committees, and rural public participation have positive roles in promoting the efficiency of rural ecological environment governance. Rural population agglomeration, financial support for agriculture and environmental protection social organizations have negative roles in the efficiency of rural ecological environment governance. From the perspective of the eastern, central, and western regions, the factors affecting the efficiency of rural ecological environment governance are different due to regional differences. Based on the above empirical analysis results, it is proposed that the key point to improve the efficiency of rural ecological environment governance in China is to promote differentiated regional coordinated governance mechanisms, described as follows.

First, balance the efficiency of rural ecological environment management between regions. The eastern and central regions need to adhere to the concept of “innovation, coordination, green, openness and sharing”, adhere to the balance between rural economic development and rural ecological environment governance, and grasp the differences in efficiency of rural ecological environment governance in the eastern, central and western regions. The eastern region should continue to maintain the leading role in rural ecological environment governance, actively mobilize the public’s enthusiasm for rural ecological environment governance, reduce the negative impact of population agglomeration on the efficiency of rural ecological environment governance, and speed up experience transmission to the central and western regions. The central region is the region with the lowest efficiency of rural ecological environment governance among the three regions. We should speed up the support of funds, talents, technology, and other resources for rural ecological environment governance, and quickly realize the high efficiency and sustainability of rural ecological environment governance. In the process of improving the efficiency of rural ecological environment governance in the western region, we should not only increase investment from the single dimension of funds, but also realize the all-round support of rural ecological environment governance, increase the enthusiasm of environmental protection organizations and the public for rural ecological environment governance, and optimize the investment factors for improving its efficiency.

Second, there is a need to explore differentiated governance based on regional reality. The resource-based rural areas in the eastern region and the central and western regions should maintain the high efficiency of rural ecological environment treatment, make rational use of the rural natural environment, realize the harmless treatment of rural domestic garbage and domestic sewage in the rural living environment, realize the ecological and livable appearance of rural villages, realize the sustainable and green development of rural production, and improve the rural production environment’s coordinated operation between the rural natural environment and rural residential environment to maximize the efficiency of rural ecological environment governance. The rural areas with a certain resource base in the central and western regions should realize the stability of rural ecological environment treatment, maintain a certain degree of effective treatment of rural domestic garbage and domestic sewage, reduce pollution and damage to the rural production environment as much as possible, and realize the significant improvement of rural ecological environment treatment. In rural areas with insufficient resources and underdeveloped economies in the central and western regions, on the premise of ensuring that the agricultural and rural farmers’ production activities are not affected, we can achieve the basic effect of rural ecological environment governance.

Third, there is a need to establish a horizontal cooperation mechanism for collaborative governance among regions. The eastern, central, and western governments should form benign and effective cooperation mechanisms, establish and improve the horizontal coordination mechanism of government collaborative governance among different parts of the eastern, central, and western regions, and form a joint force for rural ecological environment governance. Between the eastern, central, and western regions, an independent cross-regional rural ecological environment governance organization can be established, which realizes the regional overall consideration of rural ecological environment governance and has a certain organizational status. Inter-regional rural ecological environment governance issues can have unified decision-making, management, and coordination through this institution. Meanwhile, the institution can set the objectives and specific plans for rural ecological environment governance in the region to avoid “free-riding” behavior in the process of rural ecological environment governance, and establish corresponding compensation and incentive policies to maximize the enthusiasm for rural ecological environment governance in various regions. There is also a need to define the red line standard of rural ecological environment governance, and strengthen the information monitoring of resources and environment, in order to realize the overall improvement of rural ecological environment governance efficiency.

In summary, based on the multi-center governance theory and the results of empirical analysis, it is proposed that the key to improve the efficiency of China’s rural ecological environment governance is to promote a differentiated regional coordinated governance mechanism, strengthen the government’s role as a “leader” in governance, and build a multi-subject symbiotic governance structure. However, the research on the influencing factors of rural ecological environment governance efficiency in this paper is based on the analysis of spatial and temporal differences in China and draws on the existing evaluation results of ecological environment governance efficiency. In the future, the efficiency evaluation method and the verification of influencing factors can be improved, such as efficiency evaluation based on DEA and the verification of influencing factors based on the structural equation model.

## Figures and Tables

**Table 1 ijerph-19-05925-t001:** The rural ecological environment governance efficiency index system.

Types	Dimensions	Evaluating Indicators	Unit	Subjective Weight	Objective Weight	Comprehensive Weight
Output	Economic benefit	The output of green food, organic food, and pollution-free agricultural products	10^4^ ton	0.054	0.028	0.018
		Income from forestry tourism and leisure services	10^2^ million yuan	0.104	0.145	0.171
	Social benefit	Popularization rate of rural sanitary toilet penetration	%	0.059	0.043	0.028
		Popularization rate of rural tap water penetration	%	0.061	0.058	0.048
	Ecological benefit	Rate of rural greening coverage	%	0.078	0.063	0.068
Input	Rural production environment governance	Biogas treatment project for agricultural waste	10^2^ million m^3^	0.078	0.095	0.103
		Drainage area	hm^2^	0.072	0.084	0.083
		The amount of fertilizer applied	10^4^ ton	0.067	0.060	0.057
		The number of pesticides applied	10^4^ ton	0.078	0.071	0.079
	Rural ecology environment governance	Water and soil lose area	hm^2^	0.022	0.050	0.020
		Afforestation area	hm^2^	0.069	0.067	0.081
		Investment in rural landscaping construction	10^2^ million yuan	0.046	0.035	0.026
	Rural living environment governance	Biogas digester for rural domestic sewage purification	item	0.080	0.065	0.092
		Rural domestic waste transfer station	item	0.078	0.102	0.103
		Investment in rural environmental sanitation construction	10^2^ million yuan	0.054	0.035	0.024

**Table 2 ijerph-19-05925-t002:** Variable section.

Variable	Criterion Layer	Indicators	Unit
Explained variable	Rural ecological environment governance efficiency	Rural ecological environment governance efficiency	-
Explanatory variable	Rural economic development level (Re)	Actual per capita net income of rural residents	10^4^ yuan per person
	Rural population agglomeration (Rpa)	Rural population	10^3^ million person
	Size of village committee (Vc)	Number of village committee members	10^3^ person
	Financial support for agriculture (FS)	Expenditure on agriculture, forestry, and water affairs	10^4^ yuan
	Environmental protection social organization (Ep)	Number of ecological social groups	10^3^ items
	Rural public participation (Rpp)	Total number of agricultural environmental pollution and ecological damage	one item

**Table 3 ijerph-19-05925-t003:** The regression results of the Tobit model.

	China	East	Middle	West
Re	0.253 *** (0.053)	−0.325 ** (0.139)	0.527 ** (0.213)	0.731 *** (0.221)
Rpa	−1.158 *** (0.315)	−0.530 *** (0.085)	−0.395 (0.241)	−0.104 (0.082)
Vc	0.019 *** (0.006)	0.015 *** (0.005)	0.040 (0.149)	0.033 *** (0.012)
Fs	−0.141 *** (0.040)	0.312 *** (0.091)	−0.340 *** (0.115)	−0.386 *** (0.104)
Ep	−0.074 ** (0.030)	0.028 (0.025)	−0.140 * (0.080)	−0.048 (0.047)
Rpp	1.366 *** (0.190)	−0.003 ** (0.001)	0.083 ** (0.035)	−0.010 *** (0.003)
cons	1.366 (0.190)	−2.280 (0.693)	1.395 (0.755)	1.970 (0.611)

Note: ***, **, and *, respectively, represent the significance level of 1%, 5%, and 10%, and the standard deviation is in brackets.

## Data Availability

No new data were created or analyzed in this study. Data sharing is not applicable to this article.

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
