# Peer review of "What Affects Rural Ecological Environment Governance Efficiency? Evidence from China"

_ijerph, 2022, doi:10.3390/ijerph19105925_

Round 1

Reviewer 1 Report

General comments

  1. I understand the rural ecological environment governance efficiency index being a composite index (Table 1). However, it is unclear how such an index is built by authors and then used In Eq. 1 for any geographical entity (no infos available on normalization, weighting, etc., across indicators the index is composed of).
  2. Linked to the previous, most indicators should be weighted somehow (e.g. green/organic food expressed in % of total food produced rather than in absolute values), to avoid that “efficiency” depends indeed by total area size, total population, total production, etc.
  3. Equation 1 (based on Table 2) is made of quantitative indicators. Then, most of explanation of results is qualitative. In particular, section 4.2.1 should describe results based on data. Explanation should be more focused on data and coefficients rather than theoretical principles. At the current state, interpretation of results appear largely speculative.
  4. I would suggest to organize the paper by clearly distinguish three sections: analysis of results (data based); discussions (more qualitative, taking both from section 4.2.1, 4.2.2 and 5); conclusions (at the current state, very negligible)
  5. Please, make sure that all readers (including non chinese readers) can understand specific features of rural governance in China
  6. Before submitting a paper, make sure you have red it carefully. For instance, when “copy and paste” text, please check that you do it correctly (e.g. see comments below at lines 482-485, 552-556, 560-563)

Attached specific “line by line” comments (in red suggested changes in text; text in “…” is copied and pasted by the original article)

Author Response

Thank you very much for your helpful feedback and insightful comments. We have taken all of the suggestions and comments into consideration during this revision. We truly appreciate the opportunity to revise our paper, and believe that our manuscript has significantly improved in response to the ideas and recommendations of the review team.

Specific Questions:

Reviewre1#”

  1. I understand the rural ecological environment governance efficiency index being a composite index (Table 1). However, it is unclear how such an index is built by authors and then used In Eq. 1 for any geographical entity (no infos available on normalization, weighting, etc., across indicators the index is composed of).

Our Response: Data envelopment analysis is an efficiency evaluation method based on multi input and multi output to evaluate the relative effectiveness of objects. It was originally developed by American operations research scientist Charnes and others on the basis of the concept of relative effectiveness. As the efficiency of rural ecological environment governance is an evaluation that needs comprehensive indicators, referring to the research of Zhang Jianqing and others, DEA model is used to calculate the efficiency of rural ecological environment governance in China.

For the index layer in Table 1, due to the different dimensions of each index and its importance in the whole sustainable development, it is necessary to give different weights. This paper adopts the method of combining subjective and objective weights to give weights. The subjective weight is weighted by analytic hierarchy process (AHP). The subjective weight is mainly based on the research of Zhang Jianqing and others [23], and the Yaahp software is used for weighting. The objective weight is weighted by entropy method. The comprehensive weight is weighted by D-S theoretical evidence synthesis method to avoid the limitation of simple average of subjective weight and objective weight. Through the sum of the product of standardized data and weight of indicators, we can get the evaluation value of each primary indicator and incorporate the evaluation value into the DEA framework.

  1. Linked to the previous, most indicators should be weighted somehow (e.g. green/organic food expressed in % of total food produced rather than in absolute values), to avoid that “efficiency” depends indeed by total area size, total population, total production, etc.

Our Response: We appreciate your suggestion that the indicator be expressed as % in order to avoid the problem of "efficiency" depending on the local base. In this paper, when evaluating the efficiency of rural ecological environment governance, the input index is expressed by absolute value, such as soil erosion control area, investment in rural landscaping construction, rural sewage purification biogas digesters, etc. The output indicators of the three products (green food, organic food, pollution-free agricultural products) are also expressed in absolute values. In this way, when evaluating the efficiency of rural ecological environment governance in a province, it effectively solves the problem that the total area, total population and total output of the evaluation unit affect the "efficiency". It is to evaluate the efficiency of rural ecological environment governance in a region on the absolute value of input and output.

  1. Equation 1 (based on Table 2) is made of quantitative indicators. Then, most of explanation of results is qualitative. In particular, section 4.2.1 should describe results based on data. Explanation should be more focused on data and coefficients rather than theoretical principles. At the current state, interpretation of results appear largely speculative.

Our Response: We are grateful for your reminder to focus on the coefficients and data of the empirical results of Equation 1, rather than the speculative analysis alone in Section 4.2.1. In this regard, our explanation of the content arrangement in Section 4.2.1. is as follows:

Equation 1 is to use the Tobit model to empirically analyze the factors that affect the efficiency of China's rural ecological environment governance. Based on the relevant multi-center governance theories, political economy theories and previous research results, this paper focuses on the influence of the following factors including the level of rural economic development, rural population agglomeration, the scale of village committees, financial support for agriculture, environmental protection social organizations and rural public participation on the efficiency of rural ecological environment governance. These factors were selected quantitative indicators (Table 2) to reflect. After performing the Tobit model regression based on Equation 1, the empirical results are shown in Table 3.

The empirical results show that the level of rural economic development, the size of village committees, and rural public participation have positive roles in promoting the efficiency of rural ecological environment governance, and they all pass the significance test of 1%, and their action coefficients are 0.253, 0.019 and 1.366 respectively. This means that for every 1% increase in the level of rural economic development, the size of village committees, and rural public participation, the efficiency of rural ecological environment governance will increase by 0.253%, 0.019%, and 1.366%.

Besides, according to the regression results, rural population agglomeration, financial support for agriculture, environmental protection, and social organizations hurt the efficiency of rural ecological environment governance. And through the significance test of 1% or 5%, the action coefficients are -1.158, -0.141, -0.074. This means that every 1% increase in rural population agglomeration, financial support for agriculture, and environmental protection social organizations will reduce the efficiency of rural ecological environment governance by 1.158%, 0.141%, and 0.074%.

In Section 4.2., this paper has explained the coefficients and data of the empirical results of the Tobit model. In section 4.3., based on the empirical data and coefficients, this paper interprets and analyzes the factors affecting the environmental governance efficiency of China's rural ecological efficiency, which makes the content of the article more solid and clear.

  1. I would suggest to organize the paper by clearly distinguish three sections: analysis of results (data based); discussions (more qualitative, taking both from section 4.2.1, 4.2.2 and 5); conclusions (at the current state, very negligible).

Our Response: We are very grateful for your proposal to restructure the empirical analysis part of this paper, and we have adjusted the corresponding part as required. The structural adjustments are as follows:

4.2. Empirical results and analysis

4.3. Discussions

4.3.1. The analysis of Influencing Factors of national rural ecological environment governance efficiency

4.3.2. The analysis of Influencing Factors of rural ecological environment governance efficiency in the East, middle and West Regions

Thank you again for your valuable comments, which made the structure of the article clearer.

  1. Please, make sure that all readers (including non chinese readers) can understand specific features of rural governance in China.

Our Response: We are very grateful for your comments. Let us further elaborate on the concept and characteristics of China's rural ecological civilization governance, so that readers can sort out the context of the full text more clearly. Based on this, we have added relevant explanations in the introduction section.

Rural ecological governance is to build a benign interactive governance method through the participation of grassroots governments, township enterprises and the general public, and to comprehensively control the destruction of natural resources and their environment, industrial enterprise pollution, agricultural non-point source pollution and livestock and poultry breeding pollution. As well as transform and rectify the deterioration of the rural living environment, and finally achieve the goal of improving the rural ecological environment and realizing the harmonious coexistence of man and nature. The modernization of rural ecological governance is to use the power of scientific progress and environmental resources to drive rural economic and social development, and cooperate with corresponding industrial policies, fiscal policies, investment and financing policies and other means to strengthen environmental protection and ecological governance in rural areas. Ensure the coordinated development of the economy, society and ecological environment in rural areas. The core of the modernization of rural ecological governance is that the government, society, rural residents and other stakeholders can comprehensively manage the rural ecology through various methods, and effectively protect the rights and interests of rural residents while achieving rural ecological goals.

The modernization of rural ecological governance is an important part of the modernization of the national governance system and rural revitalization, a need for realizing rural modernization and development, and an important guarantee for building urban-rural integrated development. In recent years, China has explored many practical experiences in rural ecological governance: strengthen the leadership of the Communist Party of China over rural ecological governance, establish a long-term compensation mechanism for rural ecological environment, apply the latest scientific and technological achievements to rural ecological governance, enhance farmers' ecological awareness through multiple channels, advocate green agriculture and take the path of green development. As China's economic and social development has entered a new era, to realize the modernization of rural ecological governance, we must adhere to the matching of comprehensive governance and innovative construction, further strengthen the application of science and technology in ecological governance, improve the construction of laws and regulations for rural ecological civilization, and further improve the green and benign ecological environment. Circulation and other development directions, to achieve sustainable development of China's rural ecological governance modernization.

  1. Before submitting a paper, make sure you have red it carefully. For instance, when “copy and paste” text, please check that you do it correctly (e.g. see comments below at lines 482-485, 552-556, 560-563).

Our Response: We are very grateful for your attention to the formatting of the article, and when "copying and pasting" the text, we should indicate the citation and source. We carefully checked the full text, and made corrections and supplements in the original text to ensure the correctness of the full text format. Regarding the contents of lines 482-485, 552-556, and 560-563, we double-checked and determined that these contents are descriptive analysis of the empirical results of Equation 1, and the format is also satisfactory. Thank you again for your suggestion to modify the format, which makes the whole article more rigorous and clear after the revision.

Reviewer 2 Report

With rapid development, ecological environment governance still needs to be improved in China. This paper investigated the impact factors of the governance efficiency in China’s rural ecological environment with Tobit regression model. Specific comments are as follows.

[1] I don’t think the tile look good. I suggest “What Affects Rural Ecological Environment Governance Efficiency? Evidence from China”.

[2] In Abstract, why is it urgent to analyze?  I think the reasons should be mentioned.

[3] In line 45-60 “Firstly,…. Secondly,…Finally,…” I think there needs some text to connect “Firstly,…. Secondly,…Finally,…”.

[4] In line 176 “as shown in Table 1:”, I suggest “.” to replace “:”.

[5] Center table titles.

[6] In line 335 “…shown in Table 2:”, I suggest “.” to replace “:”.

[7] In line 277 “The fourth is …”and line 286 “The second is …”, these make me confused. There are six factors. I suggest session titles like 3.2.1, 3.2.2, …,3.2.6 for them or other formats to make them clearer.

Author Response

Thank you very much for your helpful feedback and insightful comments. We have taken all of the suggestions and comments into consideration during this revision. We truly appreciate the opportunity to revise our paper, and believe that our manuscript has significantly improved in response to the ideas and recommendations of the review team.

Specific Questions:

Reviewre2#”

With rapid development, ecological environment governance still needs to be improved in China. This paper investigated the impact factors of the governance efficiency in China’s rural ecological environment with Tobit regression model. Specific comments are as follows.

  1. I don’t think the tile look good. I suggest “What Affects Rural Ecological Environment Governance Efficiency? Evidence from China”.

Our Response: We are very grateful for your suggestion on replacing the title of the article. The original title of this article is "Research on the Influencing Factors of Rural Ecological Environment Governance Efficiency in China." Compared with the title of "What Affects Rural Ecological Environment Governance Efficiency? Evidence from China" proposed by you, the original

title is relatively bland and less attractive to readers, and the focus is not prominent enough. The replaced title is focused, and the question form is very good to bring readers into the content below. It fits the content and theme of this article very well. Thank you again for your suggestion on the revision of the title, we have made revisions based on the original text.

  1. In Abstract, why is it urgent to analyze? I think the reasons should be mentioned.

Our Response: In the abstract, we mentioned that it is urgent to analyze the influencing factors of rural ecological environment governance efficiency and improve China's rural ecological environment governance efficiency for rural revitalization in the new era and promote the modernization of the national environmental governance system and governance capacity. We are very grateful for your suggestion of the reason for the urgency mentioned in the abstract, we have made supplementary changes in the original text, and the specific explanation is as follows:

Since the reform and opening up, China’s economy has developed rapidly, and the level of governance has been greatly improved, which has greatly promoted the accumulation of material wealth. However, China has neglected the protection of the natural environment while consuming a large amount of natural resources. The destruction of the ecological environment has begun to endanger the sustainable survival and safety of the people, and has caused a certain degree of damage to the physical and mental health of the people and the safety of economic property. In this context, China's ecological governance modernization is particularly important.

The vast rural areas have the greatest impact on ecological governance and environmental construction, and are also the most closely connected with the ecological system. Due to the relatively backward economic development, complex natural environment and lack of ecological concepts in rural areas, it is difficult to implement ecological governance, and it is a long-term and arduous task for China's economic and social development. It must be solved well. The above is about the reason why urgency is mentioned in the abstract.

In the introduction section, this paper also expounds the main reasons for the urgent need to improve the efficiency of rural ecological environment governance and accelerate the modernization of rural ecological governance capacity and governance system: First, there is the problem of agricultural pollution emission amplification in the rural production environment. Secondly, the foundation of rural human settlements is weak and its improving speed is slow. Finally, the destruction of the rural natural environment is serious. Land desertification, deforestation, and other destructive acts are still not under control.

The above is the supplement and explanation that the efficiency of China’s rural ecological environment governance is an urgent requirement for rural revitalization in the new era and the promotion of the modernization of the national ecological governance system and governance capacity. We have supplemented it in the original text. Thank you again for your comments, the background of the article is perfected to make our article more complete and rigorous.

  1. In line 45-60 “Firstly,…. Secondly,…Finally,…” I think there needs some text to connect “Firstly,…. Secondly,…Finally,…”.

Our Response: At present, it is necessary to urgently improve the efficiency of rural ecological environment governance, and the main reasons for accelerating the modernization of rural ecological governance capabilities and governance systems are reflected in the following three aspects: due to the large amount of agricultural pollutant emissions, the ecological environment is relatively fragile and damaged and post-recovery resilience is weak. In the process of modern economic development, more and more agricultural areas have been replaced by industrialization, and the ecological environment in rural areas has been seriously damaged.

We are very grateful to you for pointing out the logical connection of the three points of the urgency to improve the efficiency of rural ecological environment governance. We have made modifications on the basis of the original text, which makes the content of our entire article more closely connected and logically strengthened. Thank you again valuable advice.

  1. In line 176 “as shown in Table 1:”, I suggest “.” to replace “:”.

Our Response: We thank you very much for pointing out the formatting issues. We have made changes based on the original text, which makes the overall context of the article clearer and more rigorous in structure.

  1. Center table titles.

Our Response: We thank you very much for pointing out the formatting problem. We have centered the title of the table on the basis of the original text, which makes the overall context of the article clearer and the structure more rigorous.

  1. In line 335 “…shown in Table 2:”, I suggest “.” to replace “:”.

Our Response: We thank you very much for pointing out the formatting problems. We have revised the punctuation marks on the basis of the original text, which makes the overall context of the article clearer and the structure more rigorous.

  1. In line 277 “The fourth is …”and line 286 “The second is …”, these make me confused. There are six factors. I suggest session titles like 3.2.1, 3.2.2, …,3.2.6 for them or other formats to make them clearer.

Our Response: We are very grateful for your suggestion that when analyzing the factors of China's rural ecological environment governance efficiency, let us divide it into 3.2.1., 3.2.2., …, 3.2.6. several parts to explain the influencing factors in an orderly and structured manner.

We have made modifications on the basis of the original text. The main purpose of this empirical study is to analyze the influencing factors that affect the efficiency of rural ecological environment governance in China, clarify the role of each influencing factor on the efficiency of rural ecological environment governance, and improve the rural ecological environment for China. Provide relevant policy recommendations on the efficiency of ecological environment governance. Based on the relevant multi-center governance theories, political economy theories and previous research results, this paper focuses on examining the influence of level of rural economic development, rural population agglomeration, the size of village committees, fiscal expenditures for agriculture, environmental protection social organizations, public participation on the efficiency of rural ecological environment governance.

Reviewer 3 Report

Rural ecological problems have been the focus of scholarly attention in recent years. This manuscript selects the evaluation indicators of rural ecological environment governance efficiency and sets explanatory variables from the perspectives of economy, population, finance, and citizen participation. Using the Tobit model, the manuscript analyzes how these variables affect rural ecological environment governance efficiency and obtain some conclusions. It is worth mentioning that the research makes some progress in the temporal and spatial scope. Nevertheless, I think there are still deficiencies in this manuscript, which makes the study immature:

Firstly, what is rural ecological environment governance efficiency? The author has defined it, but what puzzles me is why the input indicators are selected from rural production environment governance, rural ecology environment governance, and rural living environment governance. Referring to the previous research is available, but the author may also explain the logical relationship among different dimensions the author has mentioned and whether the concept is comprehensively defined.

Secondly, I noticed that the Tobit model was used in empirical analysis. What makes me curious is why not the more common Logit or Probit model? The author may explain the criteria for selecting models and the advantages of the Tobit model compared with other binary choice models.

Thirdly, the innovation of the manuscript is insufficient. The index system of rural ecological environment governance efficiency is derived from the previous research. Although the study is extended to the provincial level by obtaining data in yearbooks, unfortunately, most of the results have no new findings limited by the macro data. Even for the more abnormal results, the manuscript did not fully explain the mechanism.

In addition, since ecological differences between urban and rural areas are a long-standing and widely discussed issue, I suggest that the authors add descriptions of such phenomena to make the paper more readable, such as urban heat islands (see this recently published MDPI journal paper: Niu, L. et al. Identifying Surface Urban Heat Island Drivers and Their Spatial Heterogeneity in China's 281 Cities: An Empirical Study Based on Multiscale Geographically Weighted Regression. Remote Sens. 2021, 13, 4428. https://doi.org/10.3390/rs13214428)

L130  countermeasures, not Countermeasures

L170-176  Explain the method used to synthesize the index system, which figures out rural ecological environment governance efficiency.

L220-221 "the causal relationship" is not appropriate. The theoretical explanation of the relationship between explanatory variables and explained variables is insufficient. Thus, it is hard to describe the causal connection.

L223-224 A systematic exposition of "relevant polycentric governance theory" and "political economy theory" should be added to Literature Review.

L246 "rural population" represents whether resident population or registered population [rural population.

L255  What are the variables that measure rural population density? There is only a population scale in the manuscript, which has little representativeness of rural population aggregation?

L270-271 I have my doubts about "the continuous expansion of the scale of the village committee is conducive to providing more policy support and resource preference." That is, won't a bloated organization affect governance efficiency?

L332-333 What is "the rural population per capita"? What is the significance of taking these three indicators as core indicators?

Table2. Please redefine either the abbreviation of "Rural economic development level" or "Rural public participation", to distinguish the two clearly.

L338-339  How do indicators "the rural population per capita", "the level of rural economic protection," and "the level of rural public participation" explain "Rural population agglomeration"?

L340-345 Please modify the quantity unit.

L356-368 I noticed that there are no control variables in the model. Why not add control variables to prevent endogenous problems? Such as rural infrastructure and other variables.

L369  In the section on Empirical Results and analysis, it's better than adding descriptive statistics of variables.

L394 As far as I know, the number of village committee members in each village usually fluctuates within a small range. Hence, the number of village committee members in a province almost depends on the number of rural areas. Therefore, I have doubts about the interpretation of this result.

L433-435 Surprising about the result found. It's known that environmental governance needs financial support. Is it beneficial to environmental protection to reduce agricultural expenditure? It is suggested to provide a more convincing explanation.

L423-424 The variables selected in the model cannot embody "rural population quantity, structure, distribution, and migration." Thus, it's hard to accept what is explained.

L436-437 It's hard to understand "green water and green mountains for golden mountains and silver mountains." It is suggested to modify the translation.

L509-511  Curiously, why "the problem of policy implementation deviation" is only in the central region but not in the western area, as the west area is at a lower level of development.

L534-535  I wonder if the correlation is significant only in the middle, while it's more significant in the whole country. How to explain the latent mechanism?

L563-566  The eastern region should be more legitimate appeal channels than the middle region, which is more developed. Thus, the explanation in the manuscript makes no sense.

Author Response

Thank you very much for your helpful feedback and insightful comments. We have taken all of the suggestions and comments into consideration during this revision. We truly appreciate the opportunity to revise our paper, and believe that our manuscript has significantly improved in response to the ideas and recommendations of the review team.

Specific Questions:

Reviewre3#”

Rural ecological problems have been the focus of scholarly attention in recent years. This manuscript selects the evaluation indicators of rural ecological environment governance efficiency and sets explanatory variables from the perspectives of economy, population, finance, and citizen participation. Using the Tobit model, the manuscript analyzes how these variables affect rural ecological environment governance efficiency and obtain some conclusions. It is worth mentioning that the research makes some progress in the temporal and spatial scope. Nevertheless, I think there are still deficiencies in this manuscript, which makes the study immature:

  1. Firstly, what is rural ecological environment governance efficiency? The author has defined it, but what puzzles me is why the input indicators are selected from rural production environment governance, rural ecology environment governance, and rural living environment governance. Referring to the previous research is available, but the author may also explain the logical relationship among different dimensions the author has mentioned and whether the concept is comprehensively defined.

Our Response: We are very grateful for your suggestion to make a logical analysis and explanation of the indicator system of rural ecological environment governance efficiency, which will make this article clearer in structure and more rational and scientific in logic. The specific explanation is as follows:

Rural ecological governance is to build a benign interactive governance method through the participation of grassroots governments, township enterprises and the general public, and to comprehensively control the destruction of natural resources and their environment, industrial enterprise pollution, agricultural non-point source pollution and livestock and poultry breeding pollution. As well as transform and rectify the deterioration of the rural living environment, and finally achieve the goal of improving the rural ecological environment and realizing the harmonious coexistence of man and nature. The modernization of rural ecological governance is to use the power of scientific progress and environmental resources to drive rural economic and social development, and cooperate with corresponding industrial policies, fiscal policies, investment and financing policies and other means to strengthen environmental protection and ecological governance in rural areas. Ensure the coordinated development of the economy, society and ecological environment in rural areas.

Referring to the research of Huang and Sun, this paper constructs an input indicator system from three aspects: rural production environment governance, rural ecology environment governance, and rural living environment governance. Referring to the previous literature, the input indicators of environmental governance generally mainly include financial expenditure, pollution control investment, infrastructure investment, environmental protection scientific research and education expenditure, etc., and the output mostly focuses on the discharge of environmental pollutants. Due to the significant differences between rural and urban areas in terms of function, positioning, and industrial composition, and the degree of perfection of the rural environment-related data system is far less than that of urban areas, the constructed rural ecological environment governance index system is different from urban areas in terms of positioning and content subdivision. Based on the principles of operability, reasonable dominance, scientificity, and availability of data, the input indicators for rural ecological environment governance in the study are mainly from rural resources and environmental transformation related to rural areas, agricultural production environment governance investment and rural living environmental governance input, while output indicators are composed of three aspects: social, economic and ecological benefits.

This paper constructs an input indicator system from three aspects: rural production environment governance, rural ecology environment governance, and rural living environment governance. It includes the comprehensive consideration of economic production, ecological environment, and residents' life, which is in line with the core essence of the modernization of rural ecological environment governance, that is, the government, society, rural residents and other stakeholders can comprehensively manage rural ecology through various methods. The selection of indicators from the above three perspectives is to ensure the living needs of rural residents while taking into account the achievement of ecological environment goals and the happiness of residents' life. In summary, we believe that it is feasible, reasonable and scientific to construct an input indicator system from three aspects: rural production environment governance, rural ecology environment governance, and rural living environment governance to evaluate the efficiency of rural ecological environment governance. .

  1. Secondly, I noticed that the Tobit model was used in empirical analysis. What makes me curious is why not the more common Logit or Probit model? The author may explain the criteria for selecting models and the advantages of the Tobit model compared with other binary choice models.

Our Response: We would like to thank you very much for pointing out the scientificity and rationality of using the Tobit model to analyze the influencing factors of China's rural ecological environment governance efficiency. This will make our model construction logical and more convincing. The specific explanation is as follows:

First, let's explain the differences and connections between the Logit model, the Probit model, and the Tobit model. The Probit model is a nonlinear model that obeys a normal distribution. The simplest Probit model means that the explanatory variable Y is a 0,1 variable, and the probability of an event depends on the explanatory variable, that is, P(Y=1)=f(X), the probability of Y=1 is a function of X, where f(X) follows the standard normal distribution. The Logit model, also called the Logistic model, obeys the Logistic distribution. The Probit model follows a normal distribution. Both models are commonly used models of discrete choice models. But the Logit model is simple and straightforward and has wider application. Moreover, when the dependent variable is a nominal variable, there is no essential difference between Logit and Probit, and they can be used interchangeably in general. The difference lies in the different distribution functions used. The former assumes that the random variables obey the logistic probability distribution, while the latter assumes that the random variables obey the normal distribution. In fact, the formulas of these two distribution functions are very similar, and the difference in function values is not large. The only difference is that the tail of the logistic probability distribution function is thicker than that of the normal distribution. However, if the dependent variable is an ordinal variable, only an ordinal Probit model can be used for regression. Ordered Probit can be seen as an extension of Logit.

Tobit model refers to a type of model in which the dependent variable is roughly continuously distributed on positive values, but contains some observations with a positive probability of 0. It is also known as a censored regression model or a censored regression model, which is a type of regression with limited dependent variables. Restricted dependent variable means that the observed value of the dependent variable is continuous, but subject to certain restrictions, the obtained observed value does not fully reflect the actual state of the dependent variable.

From the research done in this paper, we firstly constructed the input-output index system of rural ecological environment governance based on the DEA model, and evaluated the efficiency of rural ecological environment governance in China by taking the cross-sectional data of rural areas in 31 provinces in 2016 as the research object. For the dependent variable rural ecological environment governance efficiency, the efficiency value of some provinces is 0, but the overall distribution is scattered in the positive range of 0-1. Therefore, this paper selects the Tobit model for demonstration.

  1. Thirdly, the innovation of the manuscript is insufficient. The index system of rural ecological environment governance efficiency is derived from the previous research. Although the study is extended to the provincial level by obtaining data in yearbooks, unfortunately, most of the results have no new findings limited by the macro data. Even for the more abnormal results, the manuscript did not fully explain the mechanism.

Our Response: We are very grateful for your suggestion to elaborate and explain the innovation of this paper in depth, which is the basis and significance of the research in this paper. The specific explanations are as follows:

After reviewing the relevant literature, we found that the previous studies mainly discussed the improvement of rural ecological environment governance efficiency from the perspective of the redundancy rate of rural ecological environment governance efficiency, and lacked to discuss the improvement of rural ecological environment governance efficiency from the perspective of its influencing factors. Therefore, this paper is devoted to analyzing the regional differences in the efficiency of China's rural ecological environment governance, exploratory identification of its influencing factors, and put forward corresponding improvement strategies, so as to promote the overall improvement of China's rural ecological environment governance efficiency.

From the perspective of constructing the efficiency index system of rural ecological environment governance, we have made improvements on the basis of previous literature. For example, in the selection of output indicators, this paper considers that the economic benefits of rural ecological environment governance output refer to the output that is conducive to enhancing economic value in the process of rural ecological environment governance. In the process of rural ecological environment governance, it is beneficial to provide a sustainable green production environment for crop production, so as to produce better ecological agricultural products and services. Based on this, this paper selects the output of green food, organic food and pollution-free agricultural products (10,000 tons) and the income of forestry tourism and leisure services (100 million yuan) as the economic benefits of the output. This is essentially different from the previous use of GDP alone to reflect economic benefits.

From the perspective of research objects, the previous literatures tend to focus on the efficiency of rural ecological environment governance in a single area or urban agglomeration. This paper expands the research object to 31 provinces in China, and explores the characteristics and problems of rural ecological environment governance in China from a macro level. On this basis, the regional differences in the efficiency of rural ecological environment governance in China are analyzed to better provide suggestions and guidance for each region according to local conditions.

From the follow-up in-depth progress, the previous research only stayed on the evaluation and analysis of the treatment efficiency of rural ecological environment, and did not further explore the influencing factors behind it. Based on this, after analyzing the regional differences in the efficiency of rural ecological environment governance in China, this paper exploratively identifies its influencing factors, uses the Tobit model to demonstrate the role of each influencing factor on the efficiency of rural ecological environment governance, and proposes corresponding measures. Improve the strategy in order to promote the overall improvement of the efficiency of China's rural ecological environment governance.

To sum up, we believe that this research has sufficient innovation, which is also the significance of the empirical exploration in this paper. Thank you again for your valuable suggestion, let us make a comprehensive exposition of the significance of this article, and make this research more scientific and reliable.

  1. In addition, since ecological differences between urban and rural areas are a long-standing and widely discussed issue, I suggest that the authors add descriptions of such phenomena to make the paper more readable, such as urban heat islands (see this recently published MDPI journal paper: Niu, L. et al. Identifying Surface Urban Heat Island Drivers and Their Spatial Heterogeneity in China's 281 Cities: An Empirical Study Based on Multiscale Geographically Weighted Regression. Remote Sens. 2021, 13, 4428. https://doi.org/10.3390/rs13214428)

Our Response: We appreciate your pointing out that the issue of ecological disparity between urban and rural areas has been brought to our attention, and adding a corresponding background note to this article makes our article more scientifically sound. We have added a background description on the ecological differences between urban and rural areas in the introduction, which is further elaborated here as follows:

The essence of the urban ecological environment problem is that the relationship between the human beings living in the city and their living environment is unbalanced. From the perspective of the boundary, the urban ecological environment problems have certain commonalities, such as the destruction of the natural environment by the urbanization process, climate change and air pollution, water pollution and so on. At the same time, as a developing country, China's ecological environment problems have its own characteristics, such as water shortage, high population density, lack of green space pollution from township industries and so on. Overall, China's urban ecological problems are mainly concentrated in the following aspects: First, the air quality is poor, and acid rain is serious, leading to a prominent "heat island" phenomenon; second, water pollution is serious; third, ground subsidence; fourth, noise pollution in general. The fifth is the continuous decline of forest coverage and the serious soil erosion; the sixth is the unreasonable utilization of land structure.

Since China's rural pollution control system has not yet been established, environmental pollution will not only rapidly turn "small pollution" into "big pollution", but also "small pollution" into "big pollution", which will bring significant negative impacts to agriculture as a disadvantaged industry and farmers as disadvantaged groups. First, the unreasonable use of pesticides brings a series of environmental problems; second, the unreasonable use of chemical fertilizers brings a series of environmental problems; third, the burning of straw aggravates air pollution and the greenhouse effect.

The prominence of rural ecological environment problems not only affects the improvement of the quality of life of urban and rural residents, but also affects the sustainable development of the entire society. In the ecological environment protection, the rural ecological environment protection should be given equal importance to promote the overall improvement of the urban and rural environmental quality. The rural ecological environment is the ecological barrier of the country, and ecological civilization must first be based on the protection and improvement of agriculture and rural ecological environment.

The urban environment and the rural environment are an organically linked whole and cannot be separated. Both rural and urban areas are in the earth's ecosystem. When there is a problem with the earth's ecosystem, it will not only affect the city, but also the countryside. For example, due to the current climate change, the ecological environment of both cities and rural areas has changed. In some places, the development of cities has polluted the rivers and soil in the countryside, and because the soil is polluted, the countryside can only provide the cities with low-quality agricultural products. Environmental problems between urban and rural areas restrict each other, which not only reduces people's quality of life, but also makes economic development unsustainable. Therefore, only by coordinating urban and rural ecological environmental protection and realizing the benign interaction between urban and rural ecology can we continuously improve people's quality of life and achieve sustainable development

  • L130 countermeasures, not Countermeasures

Our Response: We thank you very much for pointing out the inappropriate use of words in the article. We have made changes based on the original text, which makes the overall context of the article clearer and more rigorous in structure.

  • L170-176 Explain the method used to synthesize the index system, which figures out rural ecological environment governance efficiency.

Our Response: We appreciate your pointing out that we explained the rationale and method of constructing the indicator system in lines 170-176 of the article. Referring to the previous literature, this paper expands the inputs in rural ecological environment governance from three aspects: rural production environment governance investment, rural ecological environment governance investment, and rural living environment governance investment. And the output of rural ecological environment governance is developed from three aspects: economic benefits, social benefits and ecological benefits. An evaluation index system for the efficiency of rural ecological environment governance is constructed. This makes our assessment of China's rural ecological environment governance efficiency more scientific.

  • L220-221 "the causal relationship" is not appropriate. The theoretical explanation of the relationship between explanatory variables and explained variables is insufficient. Thus, it is hard to describe the causal connection.

Our Response: We thank you very much for pointing out that it is inappropriate to clarify the causal relationship between each influencing factor and the efficiency of rural ecological environment governance in the article when conducting an empirical analysis of the factors affecting the efficiency of rural ecological environment governance in China. Based on the related polycentric governance theory, political economy theory and previous research results, this paper focuses on the influence of six infect factors on the efficiency of rural ecological environment governance by constructing a Tobit model. We mainly pay attention on the role of rural economic development level, rural population agglomeration, the scale of village committees, financial support for agriculture, environmental protection social organizations and rural public participation on the efficiency of rural ecological environment governance, and do not carry out causal identification or causal inference. Therefore, we have revised the expression of "causal relationship" in the corresponding position, making the context of this article more scientific and clearer in structure.

  • L223-224 A systematic exposition of "relevant polycentric governance theory" and "political economy theory" should be added to Literature Review.

Our Response: We have added relevant references.

  • L246 "rural population" represents whether resident population or registered population [rural population.

Our Response: We would like to thank you for pointing out the meaning of “rural population”. In the context of this article, rural population refers to the population resident in rural areas, including the agricultural population and a portion of the non-agricultural population. China's current statistical system stipulates that the rural population includes: â‘  The permanent population in the number of state-owned farm households. â‘¡The resident population in the number of rural households. These include migrant workers who live in rural areas, temporary factory workers, and students who go out with their registered permanent residence in rural areas, but do not include state employees with rural registered permanent residence.

  • L255 What are the variables that measure rural population density? There is only a population scale in the manuscript, which has little representativeness of rural population aggregation?

Our Response: We appreciate your pointing out that the use of population size in this article to reflect population aggregation or density is underrepresented, the specific explanation is as follows:

We consider that the gathering of population will promote the continuous expansion of the scale of the countryside. On the one hand, the increase in the rural population will promote the rise of consumer demand, thereby accelerating the consumption of resources and the environment, and making it more difficult to improve the efficiency of rural ecological environment governance. On the other hand, the increase of the rural population is conducive to the centralized utilization of rural resources, improving the utilization efficiency of rural resources, thereby reducing the damage to the rural ecological environment. In the previous question, we mentioned the definition of rural population, and population aggregation or density refers to the number of people per unit land area. It is logically feasible and reasonable for us to use the population number to reflect the degree of population aggregation.

  • L270-271 I have my doubts about "the continuous expansion of the scale of the village committee is conducive to providing more policy support and resource preference." That is, won't a bloated organization affect governance efficiency?

Our Response: We are very grateful to you for raising the question that the size of the bloated village committee may affect the efficiency of rural ecological governance. In this regard, we also have such thoughts and questions, but as explained in this article, in general, on the one hand, he continuous expansion of the scale of the village committee is conducive to providing more policy support and resource preference for rural ecological environment governance. On the other hand, it is beneficial to call more villagers and the public to actively participate in the governance of rural ecological environment.

According to the "Organization Law of Villagers Committees of the People's Republic of China", "Village committees are composed of three to seven members, including the director, deputy directors and members. Among the members of the villagers committee, there should be women members." "Village committees are established according to the living conditions and population of the villagers, and in accordance with the principles of facilitating the self-government of the masses and conducive to economic development and social management." From this regulation, it is found that the number of village committee members in each village usually fluctuates within a small range. The number of village committee members in a province depends almost on the number of rural areas. The scale of village committees will not expand to the point of being bloated and efficient. The current impact of the scale of village committees on the efficiency of rural ecological environment governance in China should mainly be concentrated before the "inflection point."

  • L332-333 What is "the rural population per capita"? What is the significance of taking these three indicators as core indicators?

Our Response: This paper selects rural economic development level indicators, rural population agglomeration indicators, village committee size indicators, financial support for agriculture indicators, environmental protection social organization indicators and rural public participation indicators as the core explanatory variables. Among them, rural economic development level is set by the actual per capita net income of rural residents (104 yuan / person), rural population agglomeration is set by the number of rural population (100 million people) The size of the village committee is set based on the number of members of the village committee (104 people), the financial support for agriculture is set based on the expenditure on agriculture, forestry and water affairs (104 yuan), the number of ecological social groups (104) is set for environmental protection social organizations, and the total number of letters from agricultural environmental pollution and ecological damage is set for rural public participation. The efficiency of rural eco-environmental governance in various provinces and cities is selected as the explanatory variable.

  • Please redefine either the abbreviation of "Rural economic development level" or "Rural public participation", to distinguish the two clearly.

Our Response: We are very grateful for your valuable comments, reminding us to find the same abbreviations for "level of rural economic development" and "rural public participation" in Table 2. In this regard, we have re-abbreviated in Table 2, and modified all the two abbreviations that appear in the text. Thanks again for the reminder to make this article more rigorous.

  • L338-339  How do indicators "the rural population per capita", "the level of rural economic protection," and "the level of rural public participation" explain "Rural population agglomeration"?

Our Response: We uses the number of rural population (100 million people) to express the rural population agglomeration. The larger the population, the more people will gather on the unit land area.

  • L340-345 Please modify the quantity unit.

Our Response: We thank you very much for pointing out that we should revise the indicator units that affect the efficiency of rural ecological governance. We confirm and modify the corresponding units in the text again to ensure that they are completely consistent with the units of the indicators in Table 2. Thanks again for the reminder to make this article more rigorous.

  • L356-368 I noticed that there are no control variables in the model. Why not add control variables to prevent endogenous problems? Such as rural infrastructure and other variables.

Our Response: Thank you very much for your wise and visionary advice. General econometric models can add control variables. In order to study the impact of a factor on the explained variable, we must eliminate the impact of other factors on the explained variable, that is, to control the impact of other variables. By introducing other main factors affecting the explained variables into the model and estimating the model together with the factors to be studied, we can get a more accurate estimate of the factors to be studied. This article does not use control variables because it does not measure the relationship between China's rural ecological environment governance efficiency and another major variable. We calculated many factors affecting the efficiency of rural ecological environment governance in China, and took the explanatory variable as the control variable.

  • L369  In the section on Empirical Results and analysis, it's better than adding descriptive statistics of variables.

Our Response: We really appreciate your affirmation of our arrangement in section 4.2. In this part, we mainly return teachers and students to the factors that affect the efficiency of China's rural ecological environment governance in three regions, the east, the middle and the west, which makes us more confident to explain the further in-depth empirical analysis.

  • L394 As far as I know, the number of village committee members in each village usually fluctuates within a small range. Hence, the number of village committee members in a province almost depends on the number of rural areas. Therefore, I have doubts about the interpretation of this result.

Our Response: We are very grateful for your question, and believe that there is a coincidence that the size of the village committee has a positive effect on the efficiency of rural ecological environment governance. We do agree with you that the number of village committee members in each village usually fluctuates within a small range. Therefore, the number of village committee members in a province almost depends on the number of rural areas. However, since each province in China includes numerous urban administrative units, that means a large number of rural areas. Small fluctuations in the scale of village committees in a rural area are obvious at the level of the entire province, and have a significant effect on the efficiency of rural ecological environment governance. To explore whether the size of the village committee or the number of rural areas in the province has an impact on the efficiency of rural ecological environment governance, the most reliable argument is to reduce the research unit to a rural area and examine the impact of the size of the village committee on the efficiency of ecological environment governance. However, this is very difficult or even infeasible in terms of data acquisition and operability. Therefore, we can only examine the role of the scale of the village committee at the relevant macro level.

  • L433-435 Surprising about the result found. It's known that environmental governance needs financial support. Is it beneficial to environmental protection to reduce agricultural expenditure? It is suggested to provide a more convincing explanation.

Our Response: We are very grateful for your query that environmental governance requires financial support. From the empirical results made in this paper, for every 1% increase in financial support to agriculture, the efficiency of rural ecological environment governance will drop by 0.141%. We very much agree with your point of view that environmental governance needs financial support.

Referring to the research of Yuan, this paper believes that changes in local financial expenditures have three-part effects on environmental pollution: The first part is a negative direct effect. When financial expenditures increase, productive financial expenditures in polluting sectors increase and production inputs increase. Conducive to pollution treatment and reduce pollutant discharge; the second part is the negative environmental preference effect. The increase in non-productive fiscal expenditure increases the efficiency of human capital, and consumers have higher requirements for environmental quality. The third part is the pollution scale effect, the expansion of productive fiscal expenditure directly increases the capital investment of the polluting sector, which leads to the expansion of the production scale of the polluting sector, and the output and pollutant discharge increase. This results in a negative scale effect of pollution.

Since changes in local financial expenditures have both negative effects (direct effects, environmental preference effects) and positive effects (production scale effects) on environmental pollution, the impact of the expansion of local financial expenditures on environmental pollution is uncertain. It depends on the specific size of positive and negative effects.

Therefore, when this paper demonstrates the effect of fiscal support to agriculture on the efficiency of rural ecological governance, its negative effect is understandable. Due to the bias of the structure of financial support to agriculture, it has a negative effect on the pollution of rural ecological environment. For a long time, under the guidance of the development of GDP and the concept of "pollution first and then treatment", the Chinese government has made the government's fiscal expenditure structure to support agriculture to focus on production expenditures, and extensive large-scale investment to lower rural ecological environment standards. .

  • L423-424 The variables selected in the model cannot embody "rural population quantity, structure, distribution, and migration." Thus, it's hard to accept what is explained.

Our Response: We are very grateful for your questions. The specific explanation is as follows: In this paper, the number of rural population is selected to represent the level of population aggregation. This is a static indicator. It reflects the results of the dynamic evolution of the number, structure, distribution and migration of the rural population in the early stage. The subsequent explanation of the impact of population aggregation on the efficiency of rural ecological environment governance is based on the final static result of the rural population, rather than the dynamic evolution of population structure, distribution, and migration.

  • L436-437 It's hard to understand "green water and green mountains for golden mountains and silver mountains." It is suggested to modify the translation.

Our Response: We are very grateful for your suggestion, let us put it another way to say "lucid waters and lush mountains in exchange for gold and silver mountains". In fact, "exchanging lucid waters and lush mountains for gold and silver mountains" essentially means "exchange the rapid economic growth at the expense of the environment." We have revised and adjusted the corresponding parts of the text. Thank you again for your suggestions. The language expression is more concise and powerful.

  • L509-511 Curiously, why "the problem of policy implementation deviation" is only in the central region but not in the western area, as the west area is at a lower level of development.

Our Response: The effect mechanism of the size of village committee on the governance efficiency of rural ecological environment in the eastern and western regions is similar to that of the size of village committee on the governance efficiency of rural ecological environment in the whole country. The scale of the village committee has not passed the significance test on the rural ecological environment governance efficiency in the central region, which shows that there is no statistically significant correlation between the scale of the village committee and the rural ecological environment governance efficiency in the central regions.

  • L534-535  I wonder if the correlation is significant only in the middle, while it's more significant in the whole country. How to explain the latent mechanism?

Our Response: We are very grateful for your questions. The specific explanation is as follows: Environmental protection social organizations have hindered the efficiency of rural ecological environment governance in both the central region and the whole country. From the central region, it passed the 10% significance test. But nationally, it passed the 5% significance test. In this regard, we believe that the inhibitory effect of environmental protection social organizations on the efficiency of rural ecological environment governance has a spatial agglomeration effect, that is, local environmental protection social organizations will have a certain hindering effect on the efficiency of rural ecological environment governance in the territory. From the perspective of the effect coefficient, the absolute value of the coefficient in the central region is larger than the absolute value of the national coefficient, which is in line with the spatial spillover effect of environmental protection social organizations.

  • L563-566  The eastern region should be more legitimate appeal channels than the middle region, which is more developed. Thus, the explanation in the manuscript makes no sense.

Our Response: We revised the interpretation of this part. Rural public participation plays a positive role in promoting the efficiency of rural ecological environment governance in Central China. Rural public participation has a positive effect on the governance efficiency of the rural ecological environment in Central China, and the mechanism is similar to that of rural public participation in the governance efficiency of the rural ecological environment in China. Rural public participation hurts the governance efficiency of the rural ecological environment in the eastern and western regions. The appeal of the rural public in the rural ecological environment governance has not been well met, which has an impact on the scientific decision-making of rural ecological environment governance and is not conducive to the rural ecological environment governance.

Round 2

Reviewer 3 Report

The authors' revised manuscript is generally acceptable, but the references seem to be inadequate, and the following two articles, which are very relevant to this paper, are suggested as references to be added to the introduction.

About urban heat island:Niu, L. et al. Identifying Surface Urban Heat Island Drivers and Their Spatial Heterogeneity in China's 281 Cities: An Empirical Study Based on Multiscale Geographically Weighted Regression. Remote Sens. 2021, 13, 4428. https://doi.org/10.3390/rs13214428

About ural-urban dichotomy: van Vliet, Jasper, et al. "Bridging the rural-urban dichotomy in land use science." Journal of Land Use Science 15.5 (2020): 585-591.

Author Response

The specified reference has been added.

Round 3

Reviewer 3 Report

Accept in present form